# Investigating Recurrence and Eligibility Traces in Deep Q-Networks

**Jean Harb, Doina Precup**
School of Computer Science
McGill University
Montreal, QC, Canada
{jharb,dprecup}@cs.mcgill.ca

## Abstract

Eligibility traces in reinforcement learning are used as a bias-variance trade-off and can often speed up training time by propagating knowledge back over time-steps in a single update. We investigate the use of eligibility traces in combination with recurrent networks in the Atari domain. We illustrate the benefits of both recurrent nets and eligibility traces in some Atari games, and highlight also the importance of the optimization used in the training.

## 1 Introduction

Deep reinforcement learning has had many practical successes in game playing (Mnih et al. (2015),Silver et al. (2016)) and robotics (Levine & Abbeel (2014)). Our interest is in further exploring these algorithms in the context of environments with sparse rewards and partial observability. To this end, we investigate the use of two methods that are known to mitigate these problems: recurrent networks, which provide a form of memory summarizing past experiences, and eligibility traces, which allow information to propagate over multiple time steps. Eligibility traces have been shown empirically to provide faster learning (Sutton & Barto (2017), in preparation) but their use with deep RL has been limited so far (van Seijen & Sutton (2014), Hausknecht & Stone (2015)). We provide experiments in the Atari domain showing that eligibility traces boost the performance of Deep RL. We also demonstrate a surprisingly strong effect of the optimization method on the performance of the recurrent networks.

The paper is structured as follows. In Sec. 2 we provide background and notation needed for the paper. Sec. 3 describes the algorithms we use. In sec. 4 we present and discuss our experimental results. In Sec. 5 we conclude and present avenues for future work.

## 2 Background

A Markov Decision Process (MDP) consists of a tuple $\langle \mathcal{S}, \mathcal{A}, r, \mathcal{P}, \gamma \rangle$, where $\mathcal{S}$ is the set of states, $\mathcal{A}$ is the set of actions, $r : \mathcal{S} \times \mathcal{A} \mapsto \mathbb{R}$ is the reward function, $\mathcal{P}(s'|s, a)$ is the transition function (giving the next state distribution, conditioned on the current state and action), and $\gamma \in [0, 1)$ is the discount factor. Reinforcement learning (RL) (Sutton & Barto, 1998) is a framework for solving unknown MDPs, which means finding a good (or optimal) way of behaving, also called a policy. RL works by obtaining transitions from the environment and using them, in order to compute a policy that maximizes the expected return, given by $\mathbb{E}\left[ \sum_{t=0}^{\infty} \gamma^t r_t \right]$.

The state-value function for a policy $\pi : \mathcal{S} \times \mathcal{A} \to [0, 1]$, $V^\pi(s)$, is defined as the expected return obtained by starting at state $s$ and picking actions according to $\pi$. State-action values $Q(s, a)$ are similar to state values, but conditioned also on the initial action $a$. A policy can be derived from the $Q$ values by picking always the action with the best estimated value at any state.

Monte Carlo (MC) and Temporal Difference (TD) are two standard methods for updating the value function from data. In MC, an entire trajectory's return is used as the target value of the current

state.

$$\text{MC error} = \sum_{i=0}^{\infty} \gamma^i r_{t+i} - V(s_t) \tag{1}$$

In TD, the estimate of the next state's value is used to correct the current state's estimate:

$$\text{TD error} = r_t + \gamma V(s_{t+1}) - V(s_t) \tag{2}$$

Q-learning is an RL algorithm that allows an agent to learn by imagining that it will take the best possible action in the following step:

$$\text{TD error} = r_t + \gamma \max_{a'} Q(s_{t+1}, a') - Q(s_t, a_t) \tag{3}$$

This is an instance of off-policy learning, in which the agent gathers data with an exploratory policy, which randomizes the choice of action, but updates its estimates by constructing targets according to a differnet policy (in this case, the policy that is greedy with respect to the current value estimates.

## 2.1 ELIGIBILITY TRACES

Eligibility traces are a fundamental reinforcement learning mechanism which allows a trade-off between TD and MC. MC methods suffer from high variance, as many trajectories can be taken from any given state and stochasticity is often present in the MDP. TD suffers from high bias, as it updates values based on its own estimates.

Using eligibility traces allows one to design algorithms that cover the middle-ground between MC and TD. The central notion for these are $n$-step returns, which provide a way of calculating the target by using the value estimate for the state which occurs $n$ steps in the future (compared to the current state):

$$R_t^{(n)} = \sum_{i=0}^{n-1} \gamma^i r_{t+i} + \gamma^n V(s_{t+n}). \tag{4}$$

When $n$ is 1, the results is the TD target, and taking $n \to \infty$ yields the MC target.

Eligibility traces use a geometric weighting of these $n$-step returns, where the weight of the $k$-step return is $\lambda$ times the weight of the $k-1$-step return. Using a $\lambda = 0$ reduces to using TD, as all $n$-steps for $n > 1$ have a weight of 0. One of the appealing effects of using eligibility traces is that a single update allows states many steps behind a reward signal to receive credit. This propagates knowledge back at a faster rate, allowing for accelerated learning. Especially in environments where rewards are sparse and/or delayed, eligibility traces can help assign credit to past states and actions. Without traces, seeing a sparse reward will only propagate the value back by one step, which in turn needs to be sampled to send the value back a second step, and so on.

$$R_t^\lambda = (1 - \lambda) \sum_{i=0}^{\infty} \lambda^i R_t^{(i)} = (1 - \lambda) \sum_{i=1}^{\infty} \lambda^{i-1} \sum_{j=0}^{i-1} \gamma^j r_j + \gamma^{i+1} V(s_{t+i}) \tag{5}$$

This way of viewing eligibility traces is called the forward view, as states are looking ahead at the rewards received in the future. The forward view is rarely used, as it requires a state to wait for the future to unfold before calculating an update, and requires memory to store the experience. There is an equivalent view called the backward view, which allows us to calculate updates for every previous state as we take a single action. This requires no memory and lets us perform updates without having to wait for the future. However, this view has had limited success in the neural network setting as it requires using a trace on each neuron of the network, which tend to be dense and heavily used at each step resulting in noisy signals. For this reason, eligibility traces aren't heavily used when using deep learning, despite their potential benefits.

### 2.1.1 Q($\lambda$)

Q($\lambda$) is a variant of Q-learning where eligibility traces are used to calculate the TD error. As mentioned previously, the backwards view of traces is traditionally used.

A few versions of Q($\lambda$) exist, but the most used one is Watkins's Q($\lambda$). As Q-learning is off-policy, the sequence of actions used in the past trajectory used to calculate the trace might be different from the actions that the current policy might take. In that case, one should not be using the trajectory past the point where actions differ. To handle such a case, Watkins's Q($\lambda$) sets the trace to 0 if the action that the current policy would select is different from the one used in the past.

## 2.2 DEEP Q-NETWORKS

Mnih et al. (2015) introduced deep Q-networks (DQN), one of the first successful reinforcement learning algorithms that use deep learning for function approximation in a way general enough which is applicable to a variety of environments. Applying it to a set of Atari games, they used a convolutional neural network (CNN) which took as input the last four frames of the game, and output Q-values for each possible action.

Equation 6 shows the DQN cost function, where we are optimizing the $\theta$ parameters. The $\theta^-$ parameters represent frozen Q-value weights which are update at a chosen frequency.

$$\mathcal{L}(s_t, a_t|\theta) = (r_t + \gamma \max_{a'} Q(s_{t+1}, a'|\theta^-) - Q(s_t, a_t|\theta))^2 \tag{6}$$

### 2.2.1 DEEP RECURRENT Q-NETWORKS

As introduced in Hausknecht & Stone (2015), deep recurrent Q-networks (DRQN) are a modification on DQN, where single frames are passed through a CNN, which generates a feature vector that is then fed to an RNN which finally outputs Q-values. This architecture gives the agent a memory, allowing it to learn long-term temporal effects and handle partial observability, which is the case in many environments. The authors showed that randomly blanking out frames was difficult to overcome for DQN, but that DRQN learned to handle without issue.

To train DRQN, they proposed two variants of experience replay. The first was to sample entire trajectories and run the RNN from end to end. However this is very computationally demanding as some trajectories can be over 10000 steps long. The second alternative was to sample sub-trajectories instead of single transitions. This is required as the RNN needs to fill its hidden state and to allow it to understand the temporal aspect of the data.

## 2.3 OPTIMIZERS

Stochastic gradient descent (SGD) is generally the algorithm used to optimize neural networks. However, some information is lost during the process as past gradients might signal that a weight drastically needs to change, or that it is oscillating, requiring a decrease in learning rate. Adaptive SGD algorithms have been built to use this information.

RMSprop (Tieleman & Hinton (2012)), uses a geometric averaging over gradients squared, and divides the current gradient by its square root. To perform RMSprop, first we calculate the averaging as $g = \beta g + (1 - \beta)\nabla\theta^2$ and then update the parameters $\theta \leftarrow \theta + \alpha\frac{\nabla\theta}{\sqrt{g+\epsilon}}$.

DQN (Graves (2013)) introduced a variant of RMSprop where the gradient is instead divided by the standard deviation of the running average. First we calculate the running averages $m = \beta m + (1 - \beta)\nabla\theta$ and $g = \beta g + (1 - \beta)\nabla\theta^2$, and then update the parameters using $\theta \leftarrow \theta + \alpha\frac{\nabla\theta}{\sqrt{g-m^2+\epsilon}}$. In the rest of the paper, when mentioning RMSprop, we'll be referring to this version.

Finally, Kingma & Ba (2014) introduced Adam, which is essentially RMSprop coupled with Nesterov momentum, along with the running averages being corrected for bias. We have a term for the rate of momentum of each of the running averages. To calculate the update with Adam, we start with the updating the averages $m = \beta_1 m + (1 - \beta_1)\nabla\theta$, $v = \beta_2 v + (1 - \beta_2)\nabla\theta^2$, the correct their biases $\hat{m} = m/(1 - \beta_1^t)$, $\hat{v} = v/(1 - \beta_2^t)$ and finally calculate the gradient update $\theta \leftarrow \theta + \alpha\frac{\hat{m}}{\sqrt{\hat{v}+\epsilon}}$.

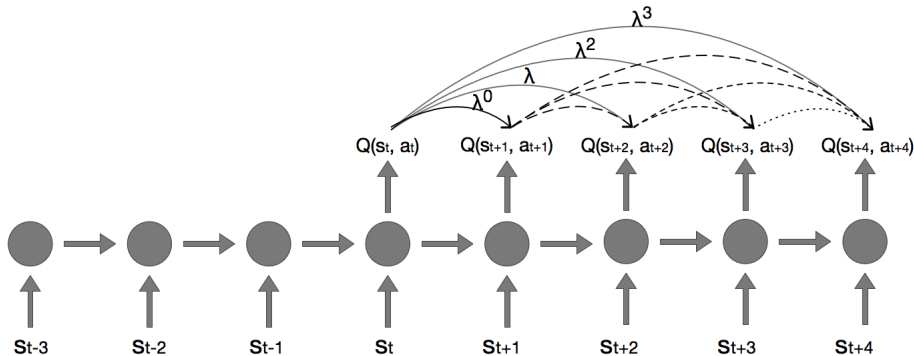

Figure 1: This graph illustrates how a sample from experience replay is used in training. We use a number of frames to fill the hidden state of the RNN. Then, for the states used for training, we have the RNN output the Q-values. Finally, we calculate each $n$-step return and weight them according to $\lambda$, where the arrows represent the forward view of each trace. All states are passed though the CNN before entering the RNN.

## 3 EXPERIMENTAL SETUP

As explained, the forward view of eligibility traces can be useful, but is computationally demanding in terms of memory and time. One must store all transitions and apply the neural network to each state in the trajectory. By using DRQN, experience replay is already part of the algorithm, which removes the memory requirement of the traces. Then, by training on sub-trajectories of data, the states must be run through the RNN with all state values as the output, which eliminates the computational cost. Finally, all that's left to use eligibility traces is simply to calculate the weighted sum of the targets, which is very cheap to do.

In this section we analyze the use of eligibility traces when training DRQN and try both RMSprop and Adam as optimizers. We only tested the algorithms on fully observable games as to compare the learning capacities without the unfair advantage of having a memory, as would be the case on partially observable environments.

### 3.1 ARCHITECTURE

We tested the algorithms on two Atari 2600 games, part of the Arcade Learning Environment (Bellemare et al. (2012)), Pong and Tennis. The architecture used is similar to the one used in Hausknecht & Stone (2015). The frames are converted to gray-scale and re-sized to 84x84. These are then fed to a CNN with the first layer being 32 8x8 filters and a stride of 4, followed by 64 4x4 filters with a stride of 2, then by 64 3x3 filters with a stride of 1. The output of the CNN is then flattened before being fed to a single dense layer of 512 output neurons, which is finally fed to an LSTM (Hochreiter & Schmidhuber (1997)) with 512 cells. We then have a last linear layer that takes the output of the recurrent layer to output the Q-values. All layers before the LSTM are activated using rectified linear units (ReLU).

As mentioned in subsection 2.2.1, we also altered experience replay to sample sub-trajectories. We use backprop through time (BPTT) to train the RNN, but only train on a sub-trajectory of experience. In runtime, the RNN will have had a large sequence of inputs in its hidden state, which can be problematic if always trained with an empty hidden state. Like in Lample & Singh Chaplot (2016), we therefore sample a slightly longer length of trajectory and use the first $m$ states to fill the hidden state. In our experiments, we selected trajectory lengths of 32, where the first 10 states are used as filler and the remaining 22 are used for the traces and TD costs. We used a batch size of 4.

All experiments using eligibility traces use $\lambda = 0.8$. Furthermore, we use Watkins's Q($\lambda$). To limit computation costs of using traces, we cut the trace off once it becomes too small. In our experiments, we choose the limit of 0.01, which allows the traces to affect 21 states ahead (when $\lambda = 0.8$). We

calculate the trace for every state in the trajectory, except for a few in the beginning, use to fill in the hidden state of the RNN.

When using RMSprop, we used a momentum of 0.95, an epsilon of 0.01 and a learning rate of 0.00025. When using Adam, we used a momentum of gradients of 0.9, a momentum of squared gradients of 0.999, an epsilon of 0.001 and a learning rate of 0.00025.

Testing phases are consistent across all models, with the score being the average over each game played during 125000 frames. We also use an $\epsilon$ of 0.05 for action selection.

---

Choose $k$ as number of trace steps and $m$ as RNN-filler steps
Initialize weights $\theta$, experience replay $\mathcal{D}$
$\theta^- \leftarrow \theta$
$s \leftarrow s_0$
**repeat**
  Initialize RNN hidden state to 0.
  **repeat**
    Choose $a$ according to $\epsilon-$greedy policy on $Q(s, a|\theta)$
    Take action $a$ in $s$, observe $s'$, $r$
    Store $s$, $a$, $r$, $s'$ in Experience Replay
    Sample 4 sub-trajectories of $m + k$ sequential transitions $(s, a, r, s')$ from $\mathcal{D}$
    $\hat{y} = \begin{cases} r & \text{s' is terminal,} \\ r + \gamma \max_{\bar{a}} Q(s', \bar{a}|\theta^-) & \text{otherwise} \end{cases}$
    **foreach** *transition sampled* **do**
      $\lambda_t = \begin{cases} \lambda & a_t = \arg\max_{\bar{a}}(s_t, \bar{a}|\theta), \\ 0 & \text{otherwise} \end{cases}$
    **end**
    **for** $l$ *from* $0$ *to* $k - 1$ **do**
      $\hat{R}_{t+l}^\lambda = \left[ \sum_{s=l}^k \left( \prod_{i=l}^s \lambda_{t+i} \right) R_{t+s}^{(s-l+1)} \right] / \left[ \sum_{s=l}^k \left( \prod_{i=l}^s \lambda_{t+i} \right) \right]$
    **end**
    Perform gradient descent on $\frac{\partial (\hat{R}^\lambda - Q(s,a|\theta))^2}{\partial \theta}$
    **Every** 10000 steps $\theta^- \leftarrow \theta$
    $s \leftarrow s'$
  **until** $s'$ *is terminal*
**until** *training complete*

**Algorithm 1:** Deep Recurrent Q-Networks with forward view eligibility traces on Atari. The eligibility traces are calculated using the $n$-step return function $R_t^{(n)}$ for time-step $t$ was described in section 2.1.

## 4 EXPERIMENTAL RESULTS

We describe experiments in two Atari games: Pong and Tennis. We chose Pong because it permits quick experimentation, and Tennis because it is one of the games that has proven difficult in all published results on Atari.

### 4.1 PONG

First, we tested an RNN model both with $\lambda = 0$ and $\lambda = 0.8$, trained with RMSprop. Figure 2 shows that the model without a trace ($\lambda = 0$) learned at the same rate as DQN, while the model with traces ($\lambda = 0.8$) learned substantially faster and with more stability, without exhibiting any epochs with depressed performance. This is probably due to the eligibility traces propagating rewards back by many steps in a single update. In Pong, when the agent hits the ball, it must wait several time-steps before the ball gets either to or past the opponent. Once this happens, the agent must assign the credit of the event back to the time when it hit the ball, and not to the actions performed after the ball had already left. The traces clearly help send this signal back faster.

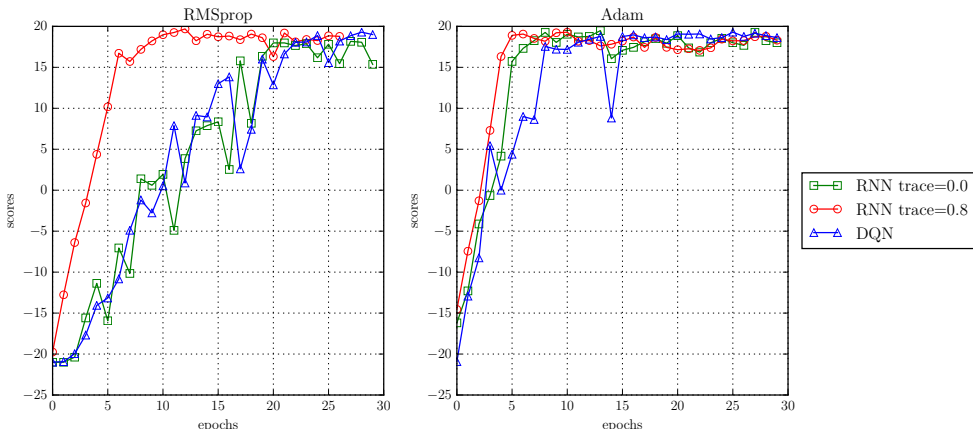

Figure 2: Test scores on Pong by training models with RMSprop vs Adam.

We then tested the same models but using Adam as the optimizer instead of RMSprop. All models learn much faster with this setting. However, the model with no trace gains significantly more than the model with the trace. Our current intuition is that some hyper-parameters, such as the frozen network's update frequency, are limiting the rate at which the model can learn. Note also that the DQN model also learns faster with Adam as the optimizer, but remains quite unstable, in comparison with the recurrent net models.

Finally, the results in Table 1 show that both using eligibility traces and Adam provide performance improvements. While training with RMSProp, the model with traces gets to near optimal performance more than twice as quickly as the other models. With Adam, the model learns to be optimal in just 6 epochs.

|  | RMSprop | Adam |
|---|---|---|
| DQN | 23 | 12 |
| RNN $\lambda = 0$ | 28 | 8 |
| RNN $\lambda = 0.8$ | 10 | 6 |

Table 1: Number of epochs before getting to 18 points in Pong. We chose 18 points as the threshold because it represents a near-optimal strategy. Testing is performed with a 5% $\epsilon$-greedy policy, stopping the agent from having a perfect score.

## 4.2 TENNIS

The second Atari 2600 game we tested was Tennis. A match consists of only one set, which is won by the player who is the first to win 6 "games" (as in regular tennis). The score ranges from 24 to -24, given as the difference between the number of balls won by the two players.

As in Pong, we first tried an RNN trained with RMSprop and the standard learning rate of 0.00025, both with and without eligibility traces (using again $\lambda = 0.8$ and $\lambda = 0$). Figure 3 shows that both RNN models learned to get optimal scores after about 50 epochs. This is in contrast with DQN, which never seems to be able to pass the 0 threshold, with large fluctuations ranging from -24 to 0. After visually inspecting the games played in the testing phase, we noticed that the DQN agent gets stuck in a loop, where it exchanges the ball with the opponent until the timer runs out. In such a case, the agent minimizes the number of points scored against, but never learns to beat the opponent. The score fluctuations depend on how few points the agent allows before entering the loop. We suspect that the agent gets stuck in this policy because the reward for trying to defeat the opponent is delayed, waiting for the ball to reach the opponent and get past it. Furthermore, the experiences of getting a point are relatively sparse. Together, it makes it difficult to propagate the reward back to the action of hitting the ball correctly.

We also notice that both the RNN with and without eligibility traces manage to learn a near-optimal policy without getting stuck in the bad policy. The RNN has the capacity of sending the signal back to past states with BPTT, allowing it to do credit assignment implicitly, which might explain their ability to escape the bad policy. Remarkably, this is the only algorithm that succeeds in getting near-optimal scores on Tennis, out of all variants of DQN (Mnih et al. (2015), Munos et al. (2016), Wang et al. (2015), Mnih et al. (2016), Schaul et al. (2015)), which tend to get stuck in the policy of delaying. The model without traces learned at a faster pace than the one with traces, arriving to a score of 20 in 45 epochs as opposed to 62 for its counterpart. It's possible that the updates for model with traces were smaller, due to the weighting of target values, indirectly leading to a lower learning rate. We also trained the models with RMSprop and a higher learning rate of 0.001. This led to the model with traces getting to 20 points in just 27 epochs, while the model without traces lost its ability to get optimal scores and never passed the 0 threshold.

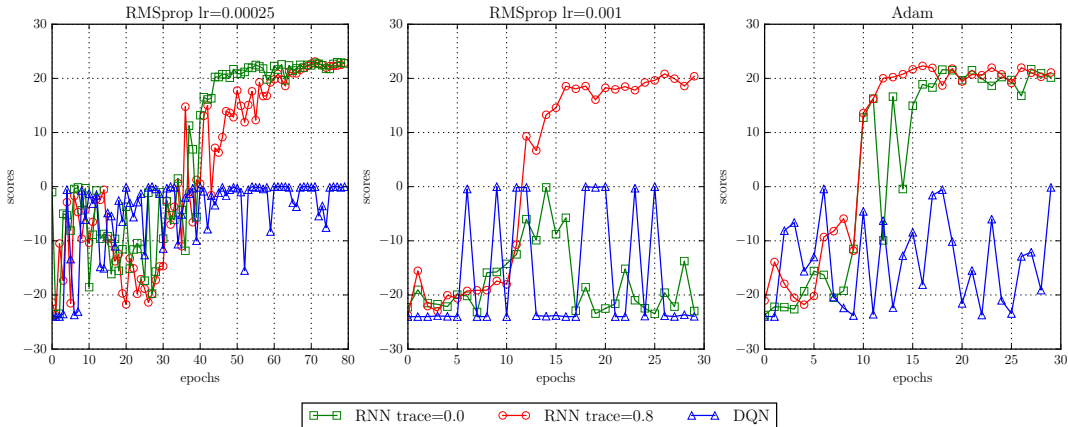

Figure 3: Test scores on Tennis comparing RMSprop and Adam.

|  | RMSprop lr=0.00025 | RMSprop lr=0.001 | Adam lr=0.00025 |
|---|---|---|---|
| DQN | N/A | N/A | N/A |
| RNN $\lambda = 0$ | 45 | N/A | 19 |
| RNN $\lambda = 0.8$ | 62 | 27 | 13 |

Table 2: Number of epochs before getting to 20 points in Tennis. N/A represents the inability to reach such a level.

We then tried using Adam as the optimizer, with the original learning rate of 0.00025. Both RNN models learned substantially faster than with RMSprop, with the RNN with traces getting to near-optimal performance in just 13 epochs. With Adam, the gradient for the positive TD is stored in the momentum part of the equation for quite some time. Once in momentum, the gradient is part of many updates, which makes it enough to overtake the safe strategy. We also notice that the model with traces was much more stable than its counterpart. The model without traces fell back to the policy of delaying the game on two occasions, after having learned to beat the opponent. Finally, we trained DQN with Adam, but the model acted the same way as DQN trained with RMSprop.

## 5 DISCUSSION AND CONCLUSION

In this paper, we analyzed the effects of using eligibility traces and different optimization functions. We showed that eligibility traces can improve and stabilize learning and using Adam can strongly accelerate learning.

As shown in the Pong results, the model using eligibility traces didn't gain much performance from using Adam. One possible cause is the frozen network. While it has a stabilizing effect in DQN, by stopping policies from drastically changing from a single update, it also stops newly learned values from being propagated back. Double DQN seems to partially go around this issue, allowing

the policy of the next state to change, while keeping the values frozen. In future experiments, we must consider eliminating or increasing the frozen network's update frequency. It would also be interesting to reduce the size of experience replay, as with increased learning speed, old observations can become too off-policy and barely be used in eligibility traces.

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
