# Peer review of "Investigating Recurrence and Eligibility Traces in Deep Q-Networks"

_ICLR 2017 — rejected_

[Official Review · AnonReviewer5 · rating 4 · confidence 5 · 16 Dec 2016]

The paper presents a deep RL with eligibility traces. The authors combine DRQN with eligibility traces for improved training. The new algorithm is evaluated on a two problems, with a single set of hyper-parameters, and compared with DQN.

The topic is very interesting. Adding eligibility traces to RL updates is not novel, but this family of the algorithms have not been explored for deep RL. The paper is written clearly, and the related literature is well-covered. More experiments would make this promising paper much stronger. As this is an investigative, experimental paper, it is crucial for it to contain a wider range of problems, different hyper-parameter settings, and comparison with vanilla DRQN, Deepmind's DQN implementation, as well as other state of the art methods.

[Official Review · AnonReviewer3 · rating 3 · confidence 5 · 16 Dec 2016]
**Interesting questions but very limited experiments**

This paper investigates the use of eligibility traces with recurrent DQN agents. As in other recent work on deep RL, the forward view of Sutton and Barto is used to make eligibility traces practical to use with neural networks. Experiments on the Atari games Pong and Tennis show that traces work better than standard Q-learning.

The paper is well written and the use of traces in deep RL is indeed underexplored, but the experiments in the paper are too limited and do not answer the most interesting questions.

As pointed out in the questions, n-step returns have been shown to work better than 1-step returns both in the classical RL literature and more recently with deep networks. [1] shows that using n-step returns in the forward view with neural networks leads to big improvements on both Atari and TORCS. Their n-step Q-learning method also combines returns of different length in expectation, while traces do this explicitly. This paper does not compare traces with n-step returns and simply shows that traces used in the forward view help on two Atari games. This is not a very significant result. It would be much more interesting to see whether traces improve on what is already known to work well with neural networks.

The other claimed contribution of the paper is showing the strong effect of optimization. As with traces, I find it hard to make any conclusions from experiments on two games with fixed hyperparameter settings. This has already been demonstrated with much more thorough experiments in other papers. One could argue that these experiments show that importance of hyperparameter values and not of the optimization algorithm itself. Without tuning the optimization hyperparameters it's hard to claim anything about the relative merits of the methods.

[1] "Asynchronous Methods for Deep Reinforcement Learning", ICML 2016.

[Official Review · AnonReviewer4 · rating 4 · confidence 4 · 17 Dec 2016]

This paper combines DRQN with eligibility traces, and also experiment with the Adam optimizer for optimizing the q-network. This direction is worth exploring, and the experiments demonstrate the benefit from using eligibility traces and Adam on two Atari games. The methods themselves are not novel. Thus, the primary contributions are (1) applying eligibility traces and Adam to DRQN and (2) the experimental evaluation. The paper is well-written and easy to understand.

The experiments provide quantitative results and detailed qualitative intuition for how and why the methods perform as they do. However, with only two Atari games in the results, it is difficult to tell how well it the method would perform more generally. Showing results on several more games and/or other domains would significantly improve the paper. Showing error bars from multiple random seeds would also improve the paper.

[Final Decision · Program Chairs · 06 Feb 2017]
**ICLR committee final decision**

The reviewers agree that the paper is clear and well-written, but all reviewers raised significant concerns about the novelty of the work, since the proposed algorithm is a combination of well-known techniques in reinforcement learning. It is worth noting that the use of eligibility traces is not very heavily explored in the deep reinforcement learning literature, but since the contribution is primarily empirical rather than conceptual and algorithmic, there is a high bar for the rigorousness of the experiments. The reviewers generally did not find the evaluation to be compelling enough in this regard. Based on this evaluation, the paper is not ready for publication.